# To Err is Humanoid; to Collaborate, Divine: A Transitional Reality Interface for Error Replay and Correction in Industrial Robotics

Lorian Marshall
*School of Engineering*
*University of Western Australia*
Perth, WA, Australia
Lorian.Marshall@uwa.edu.au

Jordan Allspaw
*Miner School of CIS*
*UMass Lowell*
Lowell, MA, USA
Jordan_Allspaw@uml.edu

Holly A. Yanco
*Miner School of CIS*
*UMass Lowell*
Lowell, MA, USA
Holly_Yanco@uml.edu

*Abstract*—**Real-time fault detection and error diagnosis are crucial to enhance trust and reduce process delays in the emerging landscape of single-human multiple-robot systems (SHMRS) within the manufacturing industry. In this paper, we propose a hybrid reality interface that utilizes real-time VR-AR transitions alongside an action sequence storage system for robot error replay. An interactive digital twin of the robotic platform, complemented by visualizations of recorded sensor data, enables operators to troubleshoot faults and adjust behaviors in an immersive environment. We also outline a future user study designed to compare this user-centric interface with traditional control methods. This work offers significant potential for advancing human-robot collaboration by facilitating a comprehensive, retrospective analysis of robot behavior.**

*Index Terms*—**Failure Detection and Recovery, Virtual Reality and Interfaces, Industrial Robots, Human-Robot Interaction**

## I. INTRODUCTION

Amidst the rapidly evolving industrial robotics landscape, robots are being deployed across a variety of applications, from assistive devices to delivery systems. This increasingly only involves human oversight for error correction and process optimization. For operators managing numerous robots, interacting with the machines can be challenging, especially when immediate attention is needed. Immersive technologies such as virtual reality (VR) and augmented reality (AR) are emerging as transformative tools for human-robot collaboration. As the manufacturing industry continues to shift from a reliance on skilled operators for basic maintenance tasks to single-human multiple-robot systems (SHMRS), there is an increasing demand for methods that optimize live error detection and troubleshooting.

In our previous work, we developed a VR interface that enabled robot operators to remotely control robots, performing tasks like navigation and dexterous manipulation [1], [2]. In this paper, we expand upon that work by introducing two key advancements: the ability to seamlessly switch between VR and AR within the same interface and the capability to record and replay robot actions, allowing operators to review past behaviors.

By integrating VR and AR into a unified, flexible framework, we offer a significant improvement over traditional diagnostic tools. This approach takes advantage of the human aptitude for 3D pattern recognition to detect errors in robot behavior, resulting in a streamlined troubleshooting process. Furthermore, the system creates a spatially accurate digital twin of the robotic platform, enabling operators to reconstruct past events and correlate them with real-time sensor data for a more comprehensive understanding of system performance.

Beyond diagnostics, this interface serves as a valuable training tool, enabling iterative simulation and validation of corrective measures prior to deployment in live environments. Corrective measures applied by the human can be used by machine learning to improve the overall system. These corrective models can also be applied across multiple robot platforms or the same type, offering potential to reduce production bottlenecks and enhance the overall reliability and safety of industrial robots.

## II. RELATED WORK

To position our work within the existing body of research, we first examine key developments in robot control interfaces. This section gives an overview of existing publications in augmented and virtual reality in robotics, transitional reality interfaces, and robot behavior replay. For a detailed perspective on the basis of this research, reading Han [3] and LeMasurier et al. [2] is recommended.

### A. Augmented Reality

Augmented reality interfaces are frequently employed in human-robot interaction (HRI) for path planning and information overlay. Increased efforts have been made to reduce the information gap in behavior prediction between the user and the robot by depicting robot characteristics and live sensor data. The gulf of evaluation model describes the inability of robots to provide indication of intent such that human users can interpret their current state [4]. Avalle et al. [5]

Lorian Marshall conducted this work while a visitor at UMass Lowell. This work was supported in part by the Office of Naval Research (N00014-23-1-2124 and N00014-23-1-2744).

developed a hardware fault management interface that overlays a highlighted digital twin of the impacted part to indicate where a fault has occurred with a virtual arrow directing the user's gaze. This adaptive model reduces the time for maintenance workers to identify and respond to common faults.

The commercialization of AR has resulted in lower manufacturing costs and improved ergonomics for extended use, making head-mounted devices (HMDs) viable for more applications. In tandem, the focus of recent projects has shifted from the historical handheld devices (HHDs) to HMDs [6]. HMDs provide flexibility for operators with their immersive views of the surrounding environment and increased awareness of nearby obstacles. In addition, they streamline the control process, negating the use of an in-hand device and mirroring natural object interactions. However, Makhataeva [7] noted that due to volume and power constraints, HMDs can be limited by latency compared to their hand-held counterparts. The user study in [8] was able to demonstrate improved task completion times and increased spatial presence with the use of an AR control interface for teleoperation.

### B. Virtual Reality

Research on VR has become a core focus of HRI for teleoperation applications. It enables remote user performance monitoring in hazardous environments [9], or of an entire network of robots in a manufacturing context [10]. Immersive VR displays have proven to be more intuitive in comparison with traditional 2D display controllers (i.e keyboard and mouse) [2]. A framework designed by Wozniak et al. [11], utilizes VR with live camera feeds for users to correct robot perception errors. VR has proven beneficial as a learning tool, by allowing the user to test responses to hypothetical scenarios by altering robot trajectories and the virtual environment without the requirement of real-world implementation [1]. This user study concluded that, despite an initial intuitive uptake exists for VR, it takes some time to overcome ingrained habits of standard systems, with participants reflecting negligible performance improvements initially relative to those who used keyboard and mouse. This observable learning curve was common to the results of the study in [12], but both hypothesized that extended training periods would return a greater difference.

### C. Transitional Reality Interfaces

Seamless transitions along the reality-virtuality continuum is a concept that has been around since the MagicBook study by Billinghurst [13], which integrated 3D spatial displays with the pages of a book. Interfaces that make use of reality transitions combine the immersive view and remote operation capabilities with the real-world reference points captured in AR. Numerous studies have identified that the major drawback of multi-modal interfaces lies in inefficient context switching between devices [14] [7]. For example, a user transitioning from VR to reality must remove the headset and controllers before using the keyboard and mouse. As a consequence, additional cognitive load is introduced that reduces an operator's

ability to perform real-time responses, and breaks concentration. Wentzel et al. [14] implemented a series of "peek-through" methods that simplify device transitions through pop-up windows to minimize hardware exchange and ocular re-immersing. The study was restricted to VR-desktop transitions, however it showed promise in extending the use case of an immersive VR environment with limited AR functionality. AR can enable the operator to review robot sensor alignment with the real-world environment, but can be restrictive when monitoring a network of robots. A new approach is therefore needed to create a holistic AR-VR interface for HRI applications in manufacturing. While some papers have investigated AR as a medium for error detection, there exists a knowledge gap in hybrid reality interfaces that incorporate transitions between VR and AR.

### D. Behavior Replay

An underexplored area of VAM-HRI is behavior replay. In a factory setting, human-robot collaboration can be challenging without effective communication of robot actions. For remote teleoperators, diagnosing an error and determining the appropriate solution can be difficult without observing it in real-time. The robot's environment may have changed since the fault occurred, making the recreation of the action sequence prone to inaccuracies. This work builds on the findings of Han [3] and Gad [15] in implementing a replay function for the operator to understand the source of the fault and correct it accordingly. It targets the two dominant manipulation and navigation errors that occurred in the user study in [3], and is also implemented in the Fetch Robotics robot. While the AR interface authored by Gad [15] was not designed for fault replay, it is relevant as it discretizes an action sequence into isolated tasks that can be accessed via a button interface. Another key review in this field is by Richards et al. [12], presents initial positive results from a user study comparing an AR behavior replay interface with a handheld tablet control scheme. This was significant as it was the first demonstration of an interface of this medium.

## III. METHODS

In our previous work, we proposed a human-in-the-loop VR interface for robot teleoperation [1], with a user study conducted to compare the VR interface against a traditional screen-based 2D medium (keyboard and mouse) to perform a series of tasks [2]. Although this prior interface supports both robot manipulation and navigation control, the improvements discussed here focus on manipulation.

### A. Design Requirements

This proposed interface is significant for showcasing the usability of behavior replay as a training mechanism for industrial robotics. It presents a novel interpretation of the Google hand-eye coordination experiment, conducted with seven robotic manipulators, to form the foundation for a large-scale data collection framework [16]. Following a similar approach, heat map visualizations projected into augmented

reality (AR) can display multiple action sequences that have been replayed. This helps inform the operator about the internal biases and tendencies of control algorithms, such as a preference for right-handed grasping motions.

The interface also shows promise in being able to establish teaching scenarios on a wider scale of other robots within the SHMRS, without the need for a fleet of physical robots such as that used in the Google study. By enabling detailed error analysis, increased understanding of the root causes of aggregate errors can be derived to assist development of mitigation strategies. Projecting this information into an AR environment through an HMD adds an interactive element, which can enhance trust in human-robot collaboration and provide insights into past behaviors and decisions.

In support of the capability of this proposed interface, three critical design requirements have been identified to justify the use case in manufacturing robotics:

1) Demonstrate real-time transitions between AR and VR.
2) Replay functionality of the past robot behavior in the action sequence where a fault occurred.
3) Visualization of the past sensor data as live outputs in AR (i.e. point cloud, camera feed, functional waypoints).

### B. Interface Architecture

The existing VR interface was migrated to Unity 6000; the most current version of Unity at the time of writing. The recent increase in support for VR features in Unity meant that a dominant portion of the previous functionality was ported from the custom code developed for the VRTK toolkit [17], to the open source OpenXR interactions toolkit for Unity Robotics.

Unity's integration of ROS communication framework in their ROS TCP Connector [18], drove us to transition from a dependence on ROS.net [19], a communication framework we developed in-house. We previously published a comparative analysis of these two frameworks [2]. Although both frameworks have their respective advantages and use-cases, the ROS TCP Connector supports ROS2, while ROS.net does not. At this time, this transitional reality interface utilizes ROS1, however compatibility with ROS2 will simplify adapting the interface for newer robots and maintaining viability for Unity.

The prototyping was performed on a Meta Quest Pro (2022) headset, with two 6DoF controllers. This headset was preferred to avoid the requirement of an external base station for processing, as programs could be built as Android apps on the device, promoting a larger workspace and increased maneuverability for the user in contrast with the HTC Vive headset in [1].

The broader OpenXR created by the Khronos group was selected over the Meta-device specific Meta OpenXR implementation, to increase the usability of the interface across a range of industry-standard VR/AR HMDs and to negate changes in the interface due to hardware. As such, features such as gaze sensing were restricted.

The interface architecture is comprised of two critical components: the Unity-based interface that is run on the HMD, and the industrial robot. Both run on the same local area network

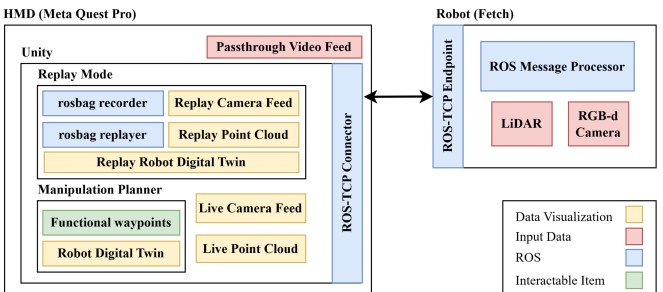

Fig. 1: A high-level diagram explaining the interface architecture. The Android app is run on the HMD, and communicates with the ROS server node on the robot through the ROS TCP Connector package. The rosbag recorder is activated when a plan is executed in the wristwatch menu goal planner and stores messages of the relevant topics. Upon entering the replay mode panel, the recording is selected, initiating the rosbag replayer. Functional gripper waypoints are spawned in as static interactable items. Replay items highlighted in yellow are topics that are monitored throughout runtime. The passthrough video feed is accessed through space setup permission on the HMD to provide the AR render.

(LAN) with socket connections to facilitate communications. This framework is illustrated in Figure 1.

Alongside the general improvements which include reliance on more open-source frameworks, we have created two major additions to the interface. The first is the ability to record and replay action sequences performed by the robot, allowing the operator to review past behaviors. The second is the ability to perform real-time reality transitions between AR and VR, such that the user can control the level of immersion of their environment. These additions used in combination can take advantage of additional context peeking in AR via seamless switching, to plan and execute a task in one reality, then change to the other reality to replay the action and review errors.

### C. Behavior Replay

The behavior replay mode considers manipulation tasks as sequences consisting of discrete actions with specific entry and exit criteria. To store these sequences, the popular rosbag tool [20] has been harnessed. Upon the robot beginning execution of a task, a ROS service call is made to snapshot the current poses of all functional gripper waypoints sent to the MoveIt planner, and a recording is initialized that publishes the live sensor data to the relevant ROS topic. This continues to record until the plan is executed and another service call is made to stop recording the sensor data. The functional waypoint poses are static elements and are sampled only once. Table I below summarizes the ROS topics that are responsible for collecting live sensor data about the robot's position, and perception of its environment. The rosbag recorder and replayer scripts were written in C++, in contrast with Python, as it is better suited to handle byte manipulation of the numerous point cloud

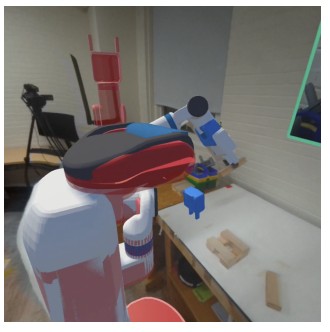
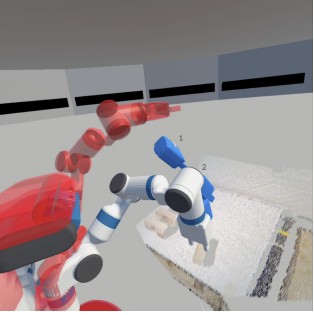

(a) in AR          (b) in VR

Fig. 2: Replay Digital Twin Visualizations, (a) in AR and (b) in VR. Both illustrate the red-tinted digital twin that is created upon loading a recorded rosbag. Blue gripper models represent the functional waypoints of the replayed task. These can be reviewed to ascertain when the error occurred. In VR (b), the recorded point cloud sensor data output can also be seen.

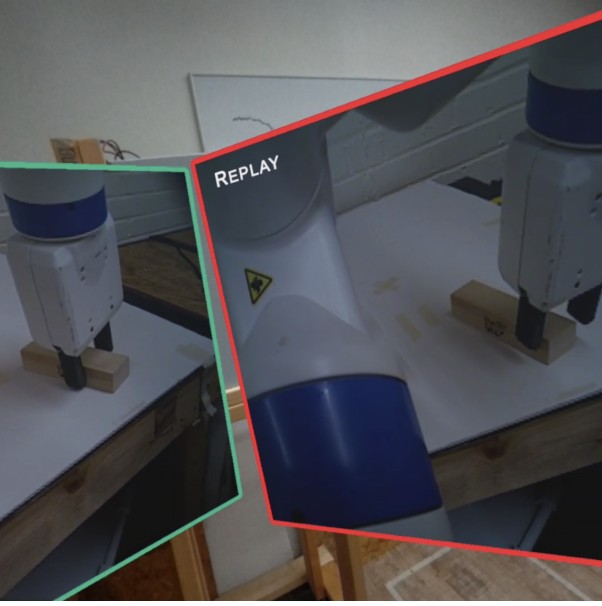

Fig. 3: Example Use of Live (green border) and Recorded Camera (red border) Feeds. The failed status of the observed pick-and-place task can be explained by the misalignment of the closed gripper position with the block seen in the pop-up 'Replay' display.

messages without timing out. The code for these scripts is available for reference at the link below[1].

This data is stored in a new rosbag that can be called. When the replay of a sequence commences, the original functional waypoints are reloaded so that the operator can visualize the intended goals of the robot. These recorded waypoints serve the same role as the normal ones created in the goal planner, and can be similarly interacted with. As a result of this, all preexisting waypoints from the planner are deleted to minimize confusion. A digital twin of the robot is also displayed that performs the actual trajectory that the robot performed. This can be differentiated from the typical twin created in the goal planner, as it has a red tint. To prevent a misrepresentation of the executed trajectory, the JointstateMsg type is used. Unlike the JointtrajectoryMsg type that is utilized by the digital twin in the manipulation goal planning mode, this message type only characterizes the position of each individual joint at the instant sampling occurs. Rather than interpolating a motion path from the series of future joint states as in the JointtrajectoryMsg, the digital twin jumps between set poses. However, as the sampling frequency is relatively high at 60fps, the human eye perceives it as smooth motion rather than individual images. The position calibrated digital twin and example functional waypoints in both reality render modes are shown in Figure 2.

The operator is able to review the saved sensor outputs and adjust a failed plan. The waypoints can be altered, repositioned, or added/deleted to create a new plan. Throughout this, the user can switch between the replay and manipulation goal planning mode before they plan and execute the new task. The naming convention for re-playable ROS topics inserts the '/replay/' tag at the beginning of the name to separate it from its predecessor.

Output sensor visualizations of the recorded point cloud and camera can be layered with the live data or toggled for

[1]https://github.com/uml-robotics/vr_bag_replay.git

comparison during the review and error inspection phase of the workflow. Since the live and recorded image quads can be observed simultaneously, a red border has been added to the replay. These visualizations offer a more comprehensive spatial awareness, giving the operator insight into how the robot interpreted what it saw and its environment. This can assist in diagnosing issues with its behavior that may not be obvious on the macroscopic scale provided by the digital twin (see Figure 3).

### D. Reality Transitions

Reality transitions between VR and AR are achieved through the native pass-through video capability on some HMDs. On the Meta Quest Pro, this is a 106° horizontal field-of-view that utilizes the suite of sensors on front facing cameras on the device to provide a live camera feed to the user. This permits the operator increased awareness of their surrounding environment and live comparisons between the digital twin and real-world robot, and negates the removal of the HMD to do so (refer to Figure 2a). The reality transition feature of the interface is reliant on the boundary being set prior to start-up. Android system permission for OpenXR at runtime is obtained through a MonoBehavior script in Unity to access data associated with the device's space setup [21]. This includes the work boundary set by the user in the front-end HMD mode that sets the height of the ground plane, and geometry of the workspace.

When the user toggles between AR and VR, controller locomotion inputs are disabled. This is done to prevent the user from attempting to manipulate the real-world in AR as if it was

TABLE I: Recorded Robot ROS Topics

| Topic Name | Purpose | Message Type |
|---|---|---|
| /base_scan | Laser range finder data | LaserscanMsg |
| /joint_states | Poses of all robot joints | JointstateMsg |
| /head_camera/depth_registered/points/filtered/throttled | Point cloud data | PointCloud2Msg |
| /head_camera/rgb/image_raw/compressed | RGB-d camera data | CompressedImageMsg |

TABLE II: Replay Interface ROS Topics

| Topic Name | Purpose | Message Type |
|---|---|---|
| /replay_duration | Duration of rosbag | TimeMsg |
| /replay_time | Current time in rosbag | TimeMsg |
| /list_of_replays | List of rosbags | StringMsg |
| /set_current_time | Sets current time in rosbag | Int32Msg |
| /set_current_replay | Sets rosbag recording | StringMsg |
| /set_replay_mode | Playback settings | StringMsg |
| /set_replay_speed | Sets speed | Float32Msg |
| /replay/gripper_goal/current | List of waypoints | PoseArrayMsg |
| /replay/functional_gripper_goal | Pose of active waypoint | JointStateMsg |

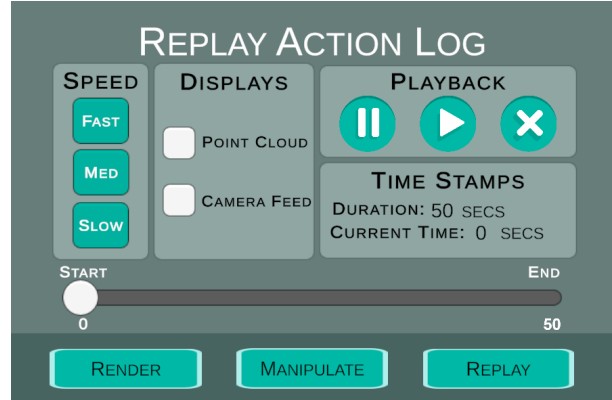

Fig. 4: Replay Mode UI Panel. The playback speed of the replay is changed by setting the delay between ROS messages. The current time in the recording can be set by dragging the scrollbar to the desired instant. The point cloud and RGB-d camera feeds are available as outputs for detailed analysis.

the VR environment, to coincide with the primary use-case of aligning the virtual world with that of the robot's surroundings. As this was a technology demonstration, the position of the character controller relative to the VR environment is carried across into AR. Hence, performing quick, successive reality transitions could introduce some cognitive dissonance if the user moves a great distance away from the robot in VR, only to teleport next to it as they activate AR.

### E. User Interactions

The user study in [2] concluded that a steeper learning curve exists for the VR interface relative to that of historical 2D control methods. As such, it was a priority to use intuitive control methods and common GUI icons where possible, to minimize added complexity. During run-time, all user interactions are accessed through a hierarchical system of wristwatch menu panels. Upon entering the program, the user is prompted to choose their dominant hand. This ensures that the wristwatch menu is attached to their non-dominant controller to avoid accidental activation when interacting with objects in the scene. Diegetic interfaces have been established in video games such as the spinal tank health bar in Dead Space as a method to reduce context switching and create a more immersive environment [22]. For this paper, a diegetic wrist-mounted menu UI was designed, that could be activated by the operator looking at their wrist as if they were viewing a watch. It was theorized that this format would be more instinctual, as watches are already associated as objects for information references in daily life. It also was thought to reduce unnecessary complexity from multiple source notifications along the peripherals of the display. The current benefits of such an interface could be extended by introducing an avatar into the virtual environment, with passive information displays like small error flag icons notifying the user that a fault has occurred without having to open the full menu. Alternatively, this passive information display could direct the user to where the robot is currently located with

an arrow, for scenarios where a network of robots may be in operation, such as a manufacturing line. To mitigate the effects of visually induced motion-sickness (VIMS) that is prevalent in extended wear use-cases of HMDs, spatial blur in stereoscopic 3D stimuli has been used [23] to help mimic the natural way human eyes perceive depth. A similar application of this has been implemented in this interface by reducing the requirement for peripheral vision focus, and hence, visual fatigue, by restricting the wristwatch panels to being head-pointing activated. Haptic feedback alerts the operator when the controllers are hovering over an interactable object with a 'bump' sensation.

A stored rosbag is reloaded in the interface by using the wristwatch menu to switch to the replay tab, and selecting a replay from the dropdown list. Once selected, the initial data is replayed, and the user can use the play, pause, stop, and speed settings to control the playback, as seen in Figure 4.

The playback is varied by changing the delay between publishing ROS messages. A central scrollbar tool that should be a familiar format to users with video playback is adopted for control of skipping to specific time stamps in the recording. This scrollbar is scaled to the duration of the recording. When dragged, it enacts an automatic pause function to prevent the replay from continuing and republishes the current time to update the current replay time respectively. Table II summarizes the ROS topics that control the replay mode functions for the interface.

## IV. IMPLEMENTATION

The Github repository with code for this interface can be referenced as seen below[2]. A demonstration video showcasing the functionality of the transitional reality interface can be found below[3].

### A. Robot Platform

A Fetch Mobile Manipulation Robot platform was chosen for implementation of the interface, as it was viable with ROS1, and made task planning comparable to past work [3] [1] [2]. Fetch is representative of the most common archetype of manufacturing robotics: articulated robots [24], and with basic traversal capabilities, can emulate most factory scenarios involving a collaborative human-robot space. Fetch has a single 7DoF arm that includes a gripper end effector. Fetch has a ground-level LiDAR for obstacle detection during navigation, and a head-mounted RGB-D camera for vision. For the purposes of this investigation, the MoveIt standard package [25] was employed for manipulation scenarios and ROS Navigation Stack for SLAM and autonomous navigation [26]. During the prototyping phase, a Gazebo-based simulated Fetch was used for testing. It should be noted that due to the nature of the physics engine in Gazebo, a minute amount of positional drifting of the Fetch model occurred over long run times, or when idle. This was not considered an issue when run on the real Fetch with the real-time calibration of visual fiducials to the AR model.

### B. Task Design

To demonstrate basic functionality of the transitional reality interface, a simple pick-and-place manipulation task was performed. The user first created a manipulation sequence for Fetch in VR that incorporated reaching for a Jenga block on the workbench, closing the gripper, picking up the block, and releasing (refer to Figure 5). The user then executed this plan to see that the robot failed to carry out a successful grasp of the block. Following this, the user would enter behavior replay mode in AR to observe the failed task, and edit the gripper degree of openness, before uploading the new plan and executing. Han [3] observed that a common failure in Fetch manipulation tasks occurred from the use of rough plywood tables as a workspace surface, which led Fetch to recognize rough features as interactable items. For the purposes of a tool investigation, errors were considered simple single-source failures. This meant that review and diagnosis could be treated as manipulation-specific. A future application of this task could include visualization of sensor outputs from multiple action sequences to view a 3D cost-map of aggregate errors. This summary of robot failures could inform future manipulation plans and feature-recognition protocols for other robots.

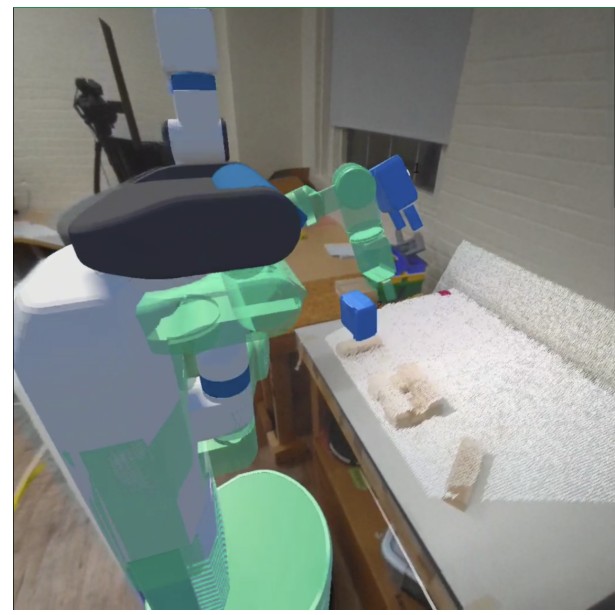

Fig. 5: Live Robot Planning Example Task in AR. The green-tinted digital twin for the manipulation planner is observed following the trajectory of a grab task defined by two gripper waypoints. The point cloud overlay is used to confirm that the robot's perception of its environment is calibrated with reality.

### C. AR Environment Alignment

Manual adjustments were implemented to combat the observable parallax between the digital twin of Fetch and the real world model in AR. A known limitation of the Meta suite of HMDs is the camera permission restrictions that block third-party developers access to the front-facing cameras. Hence, without live image data, typical calibration methods like fiducial marker processing libraries are ineffective tools for the Meta HMDs. Some methods of bypassing restrictions through screen recording may exist as suggested in [27] and require further investigation.

Fine tuning of the alignment was conducted to ensure that the digital twin pose was calibrated with that of Fetch, and the position saved. Whenever the AR render mode was entered, the camera central position in the scene was reset to the saved pose, such that the operator 'snaps' to the aligned position. This enabled the user to still utilize locomotion within VR for inspection of the robot from multiple perspectives when correcting faults that may be otherwise unobtainable due to spatially constrained environments such as warehouse aisles.

If deemed useful then future work targeting the expansion of alignment accuracy and exploration of alternative methods to AprilTags would be beneficial.

### D. Performance Considerations

Some performance considerations were made to increase the feasibility of the transitional reality interface. In contrast with our previous work on the HTC Vive, with an umbilical cord connection to a PC with a dedicated GPU, the Meta

[2]The code for this project is in a pre release version. It is accessible at https://github.com/uml-robotics/VRRInterface.git

[3]YouTube playlist: https://tinyurl.com/32f2suxb

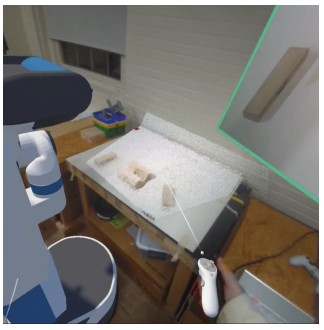 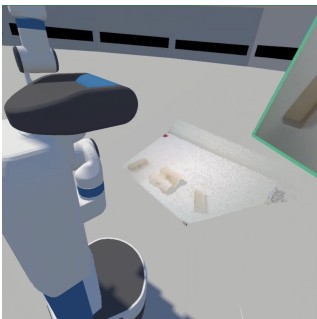

(a) in AR          (b) in VR

Fig. 6: Point Cloud Sensor Outputs. In (a), the operator is limited to viewing the robot from cleared areas but has more contextual awareness of the placement of the workspace in the real world. Increased flexibility in terms of remote teleoperation and monitoring is possible through the VR rendering (b).

Quest Pro is constrained by its graphical processing power. The compressed image camera feeds were given a threshold frame rate of 10, such that they could still give useful visual context without introducing a performance bottleneck. Similar to this, the point cloud data was down-sampled using a voxel filter to a leaf size of 0.05, and the publishing rate down to 0.5 fps. From a distance this decrease in points should be negligible to users, however it reduced the transfer rate from 110Mb/s to a more reasonable 256kb/s. A comparison of the point cloud overlay can be seen in Figure 6. Significant lag was observed when both the live and recorded point clouds were loaded in simultaneously. To prevent this problem, the UI switch was altered so that only one could be viewed at a time.

## V. FUTURE WORK

In future iterations of this work, reviewing the robot behavior prior to the action sequence where the error occurred will be beneficial, as suggested in [12], dependent on the type of error that caused the failure. Addition of functionality supporting an extended rosbag recording could prove useful for minute errors which compound over time. A collapsible mini-map is an additional feature that would assist the adaption of this interface for a scenario with a human supervisor of a network of robots. This could pinpoint the location of the specific known workspace robot which encountered an error and direct the user to that location. If, for example, the robot's path was blocked by fallen boxes in a warehouse, seeing that the robot had been stopped in a narrow aisle could help diagnose the fault without needing to move to the physical location.

At the time of writing, the rosbag recorder script tracks all plans that are executed without sorting. A future improvement is to add the rosbag dropdown selector visuals to direct attention towards sequences where an error has occurred. In combination with this, noting the time stamp in relation to the system time as part of the identification number would

help to understand the frequency and context of occurrence for common errors within the robot's operational period. More extensive work on this interface should consider transitioning to the use of ROS actions for most function monitors, saving processing power, with tasks updated as needed to the requesting node rather than as a continuous stream used in ROS topics.

A migration away from manual alignment of the virtual environment to a robust fiducial marker system that makes use of the apriltag_ros package would also be an improvement. A viable candidate for this technique would be an April Tag placed on the workspace floor, optimizing for applications that may incorporate partial occlusion and varied lighting such as that found in a warehouse. The received transform frame of the AprilTag could be applied to the simulated environment coordinate system via the AR floor plane object that is created upon completion of the work boundary setup. An alternative pathway could implement wearable calibration tags on the operator, for Fetch's onboard image sensors to recognize. Alignment of the world render should be a priority to increase the versatility of the interface in tasks that introduce navigation.

### A. Proposed User Study

While our current work demonstrates the feasibility of a transitional reality interface for error replay, an important next step is to assess how real users interact with the system. This future study could explore how different user interfaces affect task completion time and accuracy of fault diagnosis. Importantly, this will involve investigating whether a VR specific, AR specific or joint VR-AR interface streamlines the error recognition process for a simulated manufacturing human-robot collaborative environment in comparison with a traditional keyboard and mouse interface. We will collect quantitative data on the accuracy of error diagnosis and correction from a discrete set of error types, the time taken for users to recognize the error, along with qualitative data from questionnaires to determine the level of operator satisfaction and perceived ease of use for each format. Given the findings from previous studies [12] [3] [2] that a steeping learning curve exists for immersive reality interfaces, we aim to better understand with this study if the increased spatial awareness sourced from a transitional reality interface is conducive to the natural pattern recognition of humans in a 3D space and helpful to explaining past robot behaviors. We predict that both reality modes are independently limited in how they represent the workspace of the robot, but that by using seamless transitions between the two, the interface can become a promising tool for use in industry.

## VI. CONCLUSION

This novel AR-VR interface has been designed to address the challenges of fault diagnosis in manufacturing robotics. The interface architecture utilizes reality transitions with the pass-through video feed of a HMD and rosbags for retrospective analysis of manipulation tasks performed by the Fetch

robot. This transitional reality approach also opens up possibilities for broader applications across human-robot interaction research where real-time analysis and visualization of previous actions are critical as training mechanisms to inform robot behavior. Technical implementation and interaction scenarios of the interface were demonstrated and a future user study to compare its effectiveness against traditional methods has been proposed. Despite the limitations sourced from HMD permissions restrictions, autonomous error recognition and virtual environment alignment, this work presents valuable advances to the current research on robot control interfaces.

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
