# OpenReview forum: "To Err is Humanoid; to Collaborate, Divine: A Transitional Reality Interface for Error Replay and Correction in Industrial Robotics"
_humanrobotinteraction.org/HRI/2025/Workshop/VAM — HRI 2025 Workshop VAM Submission_

### Official Review · Reviewer_D9t7 · 2025-02-28

**Rating:** 7
**Confidence:** 5

**Review:**

This submission presents an investigation into a hybrid transitional reality interface for robot error replay through the implementation of a design-based methodology. The investigation into the design is grounded in a wealth of background literature and research that positions the prototype design quite well in terms of its need for development, the authors should be commended for this foundation. The findings from this study propose a future user study that better aims to understand if increased spatial awareness from a transitional reality interface is conducive to the natural pattern of human recognition in a traditional 3d space, particularly in helping explain past robot behaviours. The outputs would provide valuable insights for the VAMHRI workshop and should be included within the event.

---

### Decision · Program_Chairs · 2025-02-26

Accept